# Accuracy Assessment of Small Unmanned Aerial Vehicle for Traffic Accident Photogrammetry in the Extreme Operating Conditions of Kuwait

**Abdullah M. Almeshal** [1,*] **, Mohammad R. Alenezi** [1] **and Abdullah K. Alshatti** [2]

[1] Department of Electronic Engineering Technology, College of Technological Studies, The Public Authority for Applied Education and Training, Safat 13092, Kuwait; mr.alenezi@paaet.edu.kw

[2] Automatic Control and Systems Engineering Department, University of Sheffield, Sheffield S10 2TN, UK; alshatti82@gmail.com

* Correspondence: am.almeshal@paaet.edu.kw

**Abstract:** This study presents the first accuracy assessment of a low cost small unmanned aerial vehicle (sUAV) in reconstructing three dimensional (3D) models of traffic accidents at extreme operating environments. To date, previous studies have focused on the feasibility of adopting sUAVs in traffic accidents photogrammetry applications as well as the accuracy at normal operating conditions. In this study, 3D models of simulated accident scenes were reconstructed using a low-cost sUAV and cloud-based photogrammetry platform. Several experiments were carried out to evaluate the measurements accuracy at different flight altitudes during high temperature, low light, scattered rain and dusty high wind environments. Quantitative analyses are presented to highlight the precision range of the reconstructed traffic accident 3D model. Reported results range from highly accurate to fairly accurate represented by the root mean squared error (RMSE) range between 0.97 and 4.66 and a mean percentage absolute error (MAPE) between 1.03% and 20.2% at normal and extreme operating conditions, respectively. The findings offer an insight into the robustness and generalizability of UAV-based photogrammetry method for traffic accidents at extreme environments.

**Keywords:** UAV; aerial robotic vehicle; photogrammetry; robustness

## 1. Introduction

According to the World Health Organization (WHO), approximately 1.35 million people die each year due to road traffic accidents [1]. Accidents have been reported to be the leading cause of death for people aged between 15 and 44 years old who represent 54% of the global road accident deaths [2] and cost countries 3% approximately of the gross domestic product. In Kuwait, of a population of 4.27 million, approximately 185,000 road traffic accidents were recorded between 2016 and 2019 according to the Central Statistical Bureau (CSB) [3]. The fatalities due to road accidents were reported to be 403 per year and averaging 1.24 per day; a rate that has necessitated the implementation of rigorous traffic laws to prevent accidents. The general directorate of traffic and the ministry of interior have laid stringent traffic regulations to achieve safer roads and prevent road traffic accidents. Such regulations and frameworks will not only enable reducing the negative profound social impact on families but will contribute to enhancing the national economy of Kuwait.

The state of Kuwait, a home to high-income demographics, has an increasing number of cars that poses another challenge to the current investigation framework for accidents. Traffic congestions are very frequent during business days. This is due to the fact that the current accident investigation framework is an onsite investigation routine. In the event of a road accident, the police investigation is

carried out in two distinct phases. The first is the onsite data collection and enquiry, where police officers interrogate those involved in the incident, as well as eyewitnesses if any. The accident reconstruction diagram is sketched on a paper as part of the evidence inspection. The police then set an appointment for those involved in the accident for further investigation after analyzing the accident reconstruction sketch with experts in the second phase. The manual process of the first stage is time-consuming and inconvenient. In general scenarios, traffic accident investigation creates inconveniences in the form of increased congestions, road closures and traffic diversions due to the lengthy process of evidence collection. The unpredictable increase in the travel times lead to other possible accidents and events of road rage with traffic violations. In addition, high volumes of speeding vehicles pose threat to pedestrians when diverted into narrow neighborhood streets. The second phase of scene reconstruction is done by expert officers and usually takes few days to reconstruct the accident scene with appropriate manual calculations based on estimations of the speed of cars, the angle of impact and traffic violations. The reconstructed scene is then used as evidence for courts and insurance companies.

Considering the shortcomings of the existing methods, in terms of reconstruction accuracy and increased congestion, photogrammetry has been one of the techniques recommended in traffic accident reconstructions [4]. Photogrammetry is a method for measuring objects from photographs of the object at various angles and locations. Photogrammetry has been a popular technique in topographical assessments of geographical features—archaeological sites and civil constructions like bridges and buildings [5]. It is useful in large scale land surveys, as well as for the detection of structural defects in industrial quality control. A comparative study of the duration of investigation for different kinds of accidents reveals that photogrammetry is at least twice as fast as hand-on methods, thus, minimizing the time for road clearance [6].

With current research and development, today the reconstruction can be carried out using a range of algorithms on user-friendly software tools. An example of such tools is reported in [7] where the authors utilize a software to estimate the dynamics of the accident. The authors included a wireframe of the undamaged vehicle to measure the deformations of the vehicle by terrestrial close-range photogrammetry with cameras mounted on tripods. Aerial photography is another technique for taking photos for photogrammetry by using survey planes or unmanned aerial vehicles (UAVs). Such techniques have been actively adopted in traffic monitoring and management operations [8,9]. A wide range of payloads can be connected with commercially available small unmanned aerial vehicle (sUAV) such as thermal cameras, laser scanners, gas sensors and ground penetrating radars. Such versatility allows users to adopt sUAVs in broad applications.

## 2. Literature Review

Researchers have utilized the photogrammetry method in several applications such as 3D modelling of construction sites and buildings [10–15], stockpile estimation [16] and industrial inspection [17,18]. There is a relatively small body of literature that investigates the use of UAVs for 3D reconstruction of traffic accidents. Liu et al. [19] have conducted experiments to reconstruct a traffic accident scene using UAV. Authors have utilized referential objects placed close to the simulated accident scene as a measure of 3D model accuracy. The reported results demonstrated the efficacy of the proposed method. Limitations of the proposed method fall within adverse lighting conditions and environmental factors such as fog and rain as well as obstructions to the line of sight whilst conducting the flight mission [19]. A recent study by Pádua et al. [20] has significantly examined the performance of UAV photogrammetry of traffic accidents at various operating scenarios including the presence of barriers and obstacles and adverse lighting conditions. Authors have conducted experiments with a simulated scene surrounded by dense vegetation, trees, large canopies, electrical poles and communication aerial cables. A comparison between the actual and the measurements from the 3D model have revealed a good level of accuracy.

Another method of acquiring the data from the scene to reconstruct a 3D model is by utilizing terrestrial LiDAR scanner mounted on the UAV as reported in [21]. Accurate dimensions were obtained

within an average of 6 cm. However, this method would require a more sophisticated type of UAV and expensive equipment with interchangeable payloads and sensors. Orthophotos extracted from high definition video imagery of traffic accidents by low-cost UAVs was reported in [22]. Extracting the orthophotos allows the user to control the overlap percentages between two consecutive orthophotos to control the quality of the reconstructed 3D model of the scene. Moreover, it enables the user to accept or reject photos that may affect the quality of the 3D model of the accident scene. However, this method only considers orthophotos and authors have suggested that the inclusion of oblique images could improve the model of the accident by running multiple flight missions with different camera angles.

The research to date tends to focus on the feasibility and application of the UAV to reconstruct traffic accidents at normal operating conditions. In addition, the reported studies have been limited to developing different methods that enhance the quality of the reconstructed 3D model. Despite the importance of the application, there remains a paucity in literature on the generalizability and robustness of the method at various operating conditions and environments.

The originality of this research is that it explores the performance of low cost sUAV-based photogrammetry of traffic accidents at extreme environmental conditions of Kuwait including high temperatures, rain, dusty winds and low light operating conditions. The investigation takes the form of conducting flight experiments on a simulated traffic accident scene at three different seasons of the year. The results of this study provide an important opportunity to advance the adaptability of sUAVs in documenting traffic accidents. It is important to highlight that the approach can be generalized to any region with sub-tropical and extreme weather and is not limited to Kuwait. To the best of our knowledge, this research provides the first insight on the generalizability of using low cost sUAVs for traffic accidents at extreme operating conditions.

## 3. Methodology

### 3.1. Data Acquisition Platforms and Techniques

One of the aims of this research is to assess the performance of sUAVs in reconstructing road traffic accidents under normal and harsh operating conditions based on Structure-from-Motion technique (SfM). The SfM is a technique to acquire a series of georeferenced and overlapping 2D images from a moving camera around the object of interest to construct a 3D structure of the object. Autonomous flight planning UAV controllers made it easier to set the flight parameters required to reconstruct a 3D model of the object of interest based on the SfM technique. The captured images are processed using SfM software processing engines such as Pix4Dmapper (Pix4D SA, Lausanne, Switzerland), Metashape (Agisoft, St. Petersburg, Russia) and ArcGIS Pro (Esri, California, USA) either on local processing units or on cloud and matching features of consequent images are identified to produce point clouds in the 3D space.

The state of Kuwait is known by the subtropical desert climate with high temperatures and dust storms during summer and cold winter seasons. A careful selection of the sUAV and sensory equipment was carried out to suit the environment of Kuwait. DJI Mavic Air 2 sUAV (DJI TECHNOLOGY CO., LTD, Shenzhen, China) was found to be appropriate to conduct the proposed experiments. The selected sUAV, is a low-cost, light-weight platform that is equipped with a 48-megapixel high resolution camera on a 3-axis gimbal and hyperlight mode that suits low-lighting situations. The focal length of the camera lens impacts parameters such as the altitude of the flight and resultant clarity in the reconstructed scene output. The 3-axis gimbal allows capturing the object of interest with the required camera angles. Angles of 90 degrees and 45 degrees are commonly used in photogrammetry. A comparative analysis of cameras across a range of prices were reported in [13] and indicates that higher resolution images are better for accurate point clouds. Moreover, the sUAV can withstand a max wind resistance of Level 5 on Beaufort's scale, between 29 and 38 km/h, and has an operating temperature range between −10 °C

and 40 °C. The DJI Mavic Air 2 is also equipped with an automatic omnidirectional obstacle avoidance system that would allow operation in areas with vegetation, road lighting poles and bridges.

There are various significant parameters that ensure the quality of the photogrammetry scene reconstruction. The flight path is perhaps the most important parameter. As in Figure 1, and using SfM method, double flight paths are recommended for a complete coverage of the object of interest as well as providing overlapping between acquired images. The required percentage of lateral overlap determines the spacing between parallel flight lines. The percentages of frontal overlap, overlap in the direction of motion, and lateral overlap, overlap between consecutive parallel flights, are used to determine the speed of the sUAV as well as the path it follows. It has been reported in the literature that high frontal and lateral overlaps are recommended to improve the quality of the reconstructed scene 3D model [13].

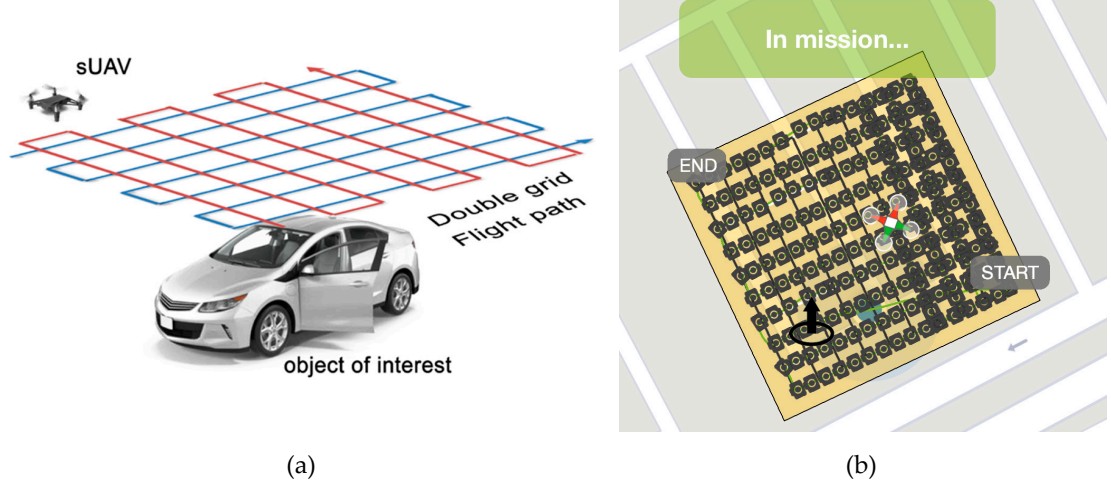

(a)                                                                (b)

**Figure 1.** (**a**) Double grid flight path over an object of interest, in this case the car (**b**) double grid flight path as executed in real-time by Pix4D software.

One significant parameter is the sUAV flight altitude from the object of interest. The flight altitude affects the ground sampling distance (GSD) which is the physical distance on the ground that corresponds to two consecutive pixel centers. Low GSD values would result in reconstructing a high-quality 3D model of the scene.

The software selection is another crucial factor to consider when the user decides to process flight data. While the scene reconstruction software needs to be user friendly and easy to learn, it is equally important to have the appropriate photogrammetry algorithms available for perusal. This is due to the fact the acquired flight data consists of georeferenced images and point clouds that need to be precisely connected with the specified frontal and lateral overlaps in order to produce a high-quality 3D model. There exist several 3D photogrammetry software packages such as ContextCapture (Bentley systems, Exton, USA), Photoscan (Agisoft, St. Petersburg, Russia), DroneDeploy (DroneDeploy, San Fransisco, USA) and Pix4Dcapture (Pix4D, Lausanne, Switzerland). However, these software packages have major differences in terms of cloud processing, superiority in terms of accuracy and dealing with outlier point clouds, and integration with UAVs and controllers. ContextCapture, Arc3D and Photoscan provide various functionalities with cloud processing of input images from the user but lacks flight path planning application for proper 3d modelling process. DroneDeploy and Pix4Dcapture have flight path planning applications to control various UAVs as well as cloud processing of the images. However, a major drawback of DroneDeploy is that their flight planning app is not supported to be downloaded in all countries. On the other hand, Pix4Dcapture flight planning application is a more generalized software solution.

In terms of accuracy comparison, a recent study reported a comprehensive comparison several photogrammetry software and have shown that Pix4D results in accurate and comparable results with the ground-truth model with less than 1.5 ground sampling distance (1.5 GSD) at worst [23].

The choice between different licensing and open source software packages is another consideration for budgetary constraints. Pix4Dcapture was selected for flight path planning, data acquisition and 3D model reconstruction due to its compatibility with DJI Mavic Air 2, license availability and the aforementioned terms of accuracy and generalizability in reconstructing the 3D model of the accident scene.

Different workflow approaches were reported in the literature where each step of data acquisition, flight planning, 3D scene reconstruction, visualization and analyses was performed using different software packages [24]. In this research, the use of Pix4D would provide a comprehensive workflow that includes, flight path planning, data management, cloud processing, 3D scene reconstruction and analysis rather than using multiple workflows. These functions would provide the user with the ability to have an in-situ 3D reconstruction and visualization.

*3.2. Experiments and Methods*

Investigating the measurement's accuracy of sUAV photogrammetry in road accidents is a primary concern to adopting the technology. To investigate the performance and accuracy of the proposed photogrammetry framework in real-life scenarios, three sets of experimental scenarios were conducted. The scenarios are categorized as:

1. Scenario 1: Accuracy assessment with variation of flight altitude;
2. Scenario 2: Accuracy assessment with variation of frontal and lateral overlaps;
3. Scenario 3: Accuracy assessment at extreme operating conditions.

The first two experiments were conducted using a simulated accident scene with the presence of trees, obstacles and road light poles. While in the third scenario, two experiments were conducted at extreme operating conditions of extreme temperatures, light intensities, and environmental factors. Figure 2 illustrates the experiment flowchart.

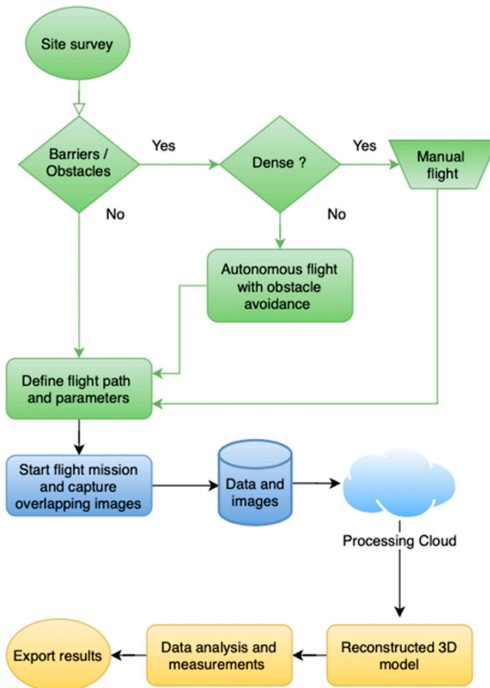

**Figure 2.** Flowchart of small unmanned aerial vehicle (sUAV) photogrammetry in traffic accident investigation.

In photogrammetry, line-of-sight (LOS) vision between the sUAV and the object under study is essential for detailed and accurate reconstruction of the 3D model. The experiment begins with an assessment of the area for the presence of possible obstacles, dense vegetations and barriers to determine the level of manual intervention required in maneuvering the sUAV. Once the area is surveyed, flight parameters, such as flight altitude, path shape and length, lateral and frontal overlaps, are then defined for the flight mission. Pix4Dcapture software enables autonomous flight missions and provides estimated flight time that is calculated based on the defined mission parameters.

The acquired images are then uploaded to the processing cloud to reconstruct the 3D model of the accident scene. The photogrammetric processing of the images starts with filtering out-of-focus images and inspecting the georeferenced information and overlapping each image. Images are then mapped into their corresponding locations in the 3D space. The model is then reconstructed using point clouds or as a mesh model that can be analyzed and measured within the Pix4D software environment.

The first scenario investigates the implications of flight altitude and GSD on the accuracy of the measurements of the road accident scene. In addition, the second scenario examines the relationship between the overlapping percentages and the quality of the 3D constructed model in terms of the accuracy of measurements.

The third scenario investigates the performance of the sUAV photogrammetry system at various operating conditions. Two sub-experiments are defined in this scenario to investigate the performance of the proposed framework in the extreme operating conditions of Kuwait at different seasons. Table 1 presents the details of the experimental scenarios. Experiment A presents extreme operating conditions during summer in Kuwait with high temperature and dust wind gusts, while Experiment B presents extreme operating conditions during winter in Kuwait with scattered rain and low light.

**Table 1.** Experimental scenarios.

| Experiment | Month | Time | Temperature Range | Wind Speed | Environmental Factors | Notes |
|:---:|:---:|:---:|:---:|:---:|:---:|:---:|
| A | July | 13:00 | 47 °C | 25 kph | Rising dust | High temperature Daylight high wind with dust gusts |
| B | November | 19:00 | 8 °C | 15 kph | Scattered rain | low temperature, low light, medium wind |

Based on the experiment design, various flight missions were conducted throughout the year to assess the performance of the proposed method at various operating conditions. The next section presents the results and analyses of each experiments.

## 4. Results and Analysis

### 4.1. Scenario 1: Variable Flight Altitude and GSDs

A simulated accident scene in a car parking space is used for the first scenario to investigate the effect of different flight altitudes and GSDs on the accuracy of the measurements of the accident scene. The area has some trees, four other cars of various sizes and road light poles as shown in Figure 3. There were no obstructions to the line-of-sight of the sUAV during the flight. Manual tape measurements of the accident scene were carried out and presented in Table 2. A set of 17 segments were identified in this scene, seven of which are associated with the physical features of Car A, and B, and three are associated with the spacing between them. A mannual measurement process was conducted within approximately 30 min.

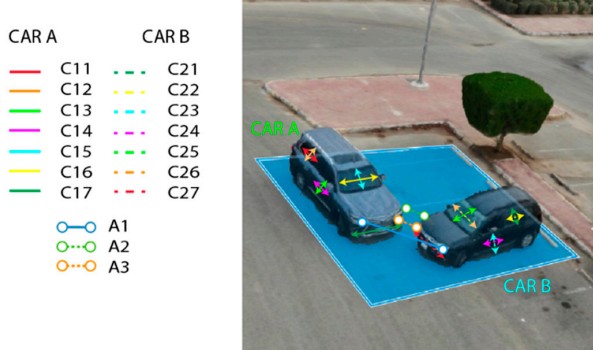

**Figure 3.** Scenario 1 Simulated accident scene.

**Table 2.** Measured segments.

| Car A | Car B | Between Car A and Car B |
|---|---|---|
| Length of the back window (C11) | Length of the back window (C21) | |
| Width of the back window (C12) | Width of the back window (C22) | Distance between the farthest |
| Length of the front door (C13) | Length of the front door (C23) | front lights (A1) |
| Width of the front door (C14) | Width of the front door (C24) | Distance from tire to tire (A2) |
| Length of the front mirror (C15) | Length of the front mirror (C25) | Distance from bumper to bumper |
| Width of the front mirror (C16) | Width of the front mirror (C26) | (A3) |
| Width of the bumper (C17) | Width of the bumper (C27) | |

Figure 4 presents the orthomosaic and the sparse digital surface model (DSM) of the scene. Four flight missions were conducted at different heights. Flight altitudes were set at 17 m, 25 m, 30 m and 40 m that correspond to GSD values of 0.17 cm/pixel, 0.72 cm/pixel, 1.18 cm/pixel and 1.52 cm/pixel, respectively. Several flight patterns exist such as circular, polygon, grid and double grid flight paths. The polygon and grid flight paths are used mainly for producing 2D map outputs of the scene. Whereas in circular and double grid flight paths, the acquired point clouds would be used to reconstruct the 3D model of the scene. However, circular flight pattern does not provide the user the ability to define the percentage overlap of the images. On the other hand, in double grid flight pattern the user can define the percentage of the lateral and frontal overlap of the images to control the quality of the reconstructed 3D model of the scene. The sUAV path was defined as a double grid path to provide the required frontal and lateral overlapping percentage of 90% equally. Table 3 presents the experiment parameters for each flight mission including height, GSD, camera angle and flight time of each experiment.

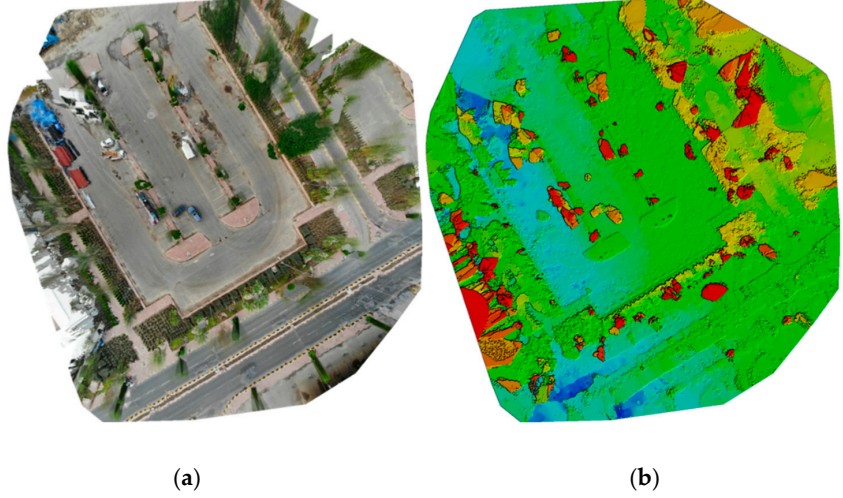

(**a**)             (**b**)

**Figure 4.** (**a**) Orthomosaic and (**b**) digital surface model (DSM) images of the scene.

**Table 3.** Scenario 1 flight mission parameters.

| Flight Mission | Height (in m) | GSD (in cm/pixel) | Camera Angle (Degrees) | Total Flight Time (Minutes) |
| --- | --- | --- | --- | --- |
| 1 | 17 | 0.17 | 45° | 8:32 min |
| 2 | 25 | 0.72 | 45° | 7:52 min |
| 3 | 30 | 1.18 | 30° | 6:12 min |
| 4 | 40 | 1.52 | 20° | 4:09 min |

The flight missions were conducted at different flight altitudes and data were uploaded and analyzed in Pix4D cloud platform. Figure 5 illustrates the 3D reconstructed model of the accident scene in Pix4D cloud platform. Pix4D provides various tools to measure distances and estimate areas from the 3D reconstructed mesh in the scene. Four different 3D models were reconstructed from each flight mission to represent a certain altitude. The tools are utilized to measure the segments of each model and the measurements are compared with the manual measurement to assess the accuracy of the photogrammetry process at different altitudes. Table 4 presents the manual measurement and the sUAV recorded measurement extracted from the 3D reconstructed model of the accident scene at each altitude. It can be noted that the lower the altitude of the sUAV the lower the measurement error. This can be explained due to the fact that lower altitude would result in a smaller GSD that allows for a more accurate and detailed 3D model reconstruction process. Higher resolution 3D model enables the user to define the points of interest with precision using the measurement tools of the Pix4D cloud platform.

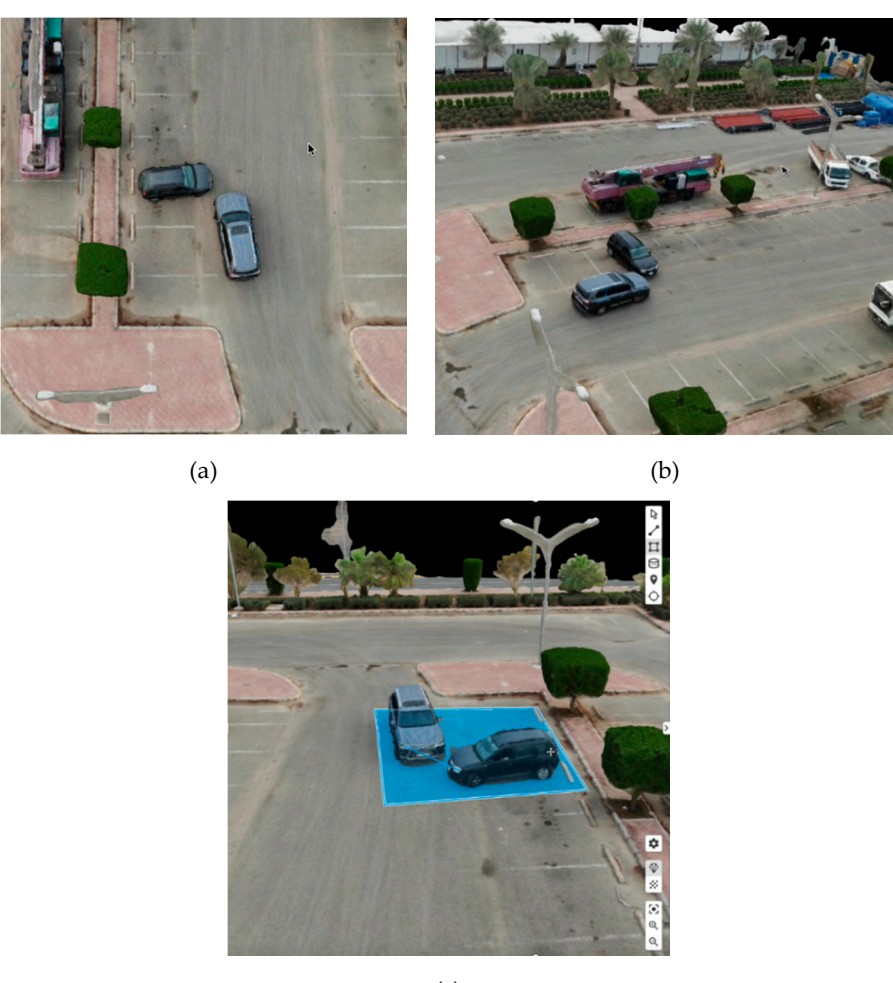

(a)　　　　　　　　　　　　　　　　　(b)

(c)

**Figure 5.** 3D reconstructed model of the accident scene of scenario 1 (**a**) top view (**b**) isometric view (**c**) measuring tools of the Pix4D cloud platform.

**Table 4.** Manual measurements vs. sUAV measurements of the accident scene at various flight altitudes.

| Segment | Manual Measurements | Altitude = 17 m | | Altitude = 25 m | | Altitude = 30 m | | Altitude = 40 m | |
|---|---|---|---|---|---|---|---|---|---|
| | | sUAV (cm) | Error % | sUAV (cm) | Error % | sUAV (cm) | Error % | sUAV (cm) | Error % |
| C11 | 85.00 | 86.45 | **1.71** | 83.23 | 2.08 | 87.36 | 2.78 | 80.97 | 4.74 |
| C12 | 48.00 | 48.95 | **1.98** | 46.18 | 3.79 | 45.39 | 5.44 | 51.34 | 6.96 |
| C13 | 108.00 | 109.17 | **1.08** | 106.48 | 1.41 | 105.49 | 2.32 | 111.12 | 2.89 |
| C14 | 79.00 | 79.65 | **0.82** | 77.16 | 2.33 | 75.12 | 4.91 | 82.49 | 4.42 |
| C15 | 155.00 | 155.46 | **0.30** | 153.17 | 1.18 | 157.54 | 1.64 | 158.73 | 2.41 |
| C16 | 75.00 | 75.55 | **0.73** | 73.48 | 2.03 | 72.08 | 3.89 | 70.64 | 5.81 |
| C17 | 114.00 | 115.37 | **1.20** | 112.06 | 1.70 | 111.39 | 2.29 | 117.6 | 3.16 |
| C21 | 60.00 | 61.35 | **2.25** | 58.44 | 2.60 | 57.03 | 4.95 | 63.78 | 6.30 |
| C22 | 42.00 | 42.89 | **2.12** | 40.06 | 4.62 | 39.16 | 6.76 | 37.19 | 11.45 |
| C23 | 105.00 | 106.06 | **1.01** | 103.43 | 1.50 | 101.25 | 3.57 | 107.87 | 2.73 |
| C24 | 125.00 | 126.09 | **0.87** | 123.37 | 1.30 | 121.13 | 3.10 | 128.58 | 2.86 |
| C25 | 121.00 | 122.02 | **0.84** | 119.04 | 1.62 | 117.88 | 2.58 | 123.96 | 2.45 |
| C26 | 84.00 | 84.45 | **0.54** | 82.25 | 2.08 | 81.86 | 2.55 | 87.72 | 4.43 |
| C27 | 152.00 | 153.04 | **0.68** | 150.4 | 1.05 | 155.96 | 2.61 | 147.94 | 2.67 |
| A1 | 289.00 | 290.12 | **0.39** | 287.28 | 0.60 | 286.51 | 0.86 | 284.46 | 1.57 |
| A2 | 133.00 | 133.53 | **0.40** | 131.32 | 1.26 | 130.4 | 1.95 | 135.73 | 2.05 |
| A3 | 95.00 | 94.5 | **0.53** | 93.26 | 1.83 | 98.38 | 3.56 | 98.44 | 3.62 |

Further statistical tests were conducted to evaluate the precision of sUAV measurements. Four performance indicators were calculated for each flight mission to assess the accuracy such as mean square error (MSE), root mean square error (RMSE), coefficient of variation of root mean square error (CVRMSE) and the mean absolute percentage error (MAPE). Table 5 provides a comparison of the four accuracy indicators of all the flight missions of scenario 1. The RMSE of the measurements range from 0.97 to 3.70 with a MAPE between 1.03% and 4.15% that correspond to lowest and highest flight altitudes, respectively. It can be observed that flight altitude of 17 m above the accident scene has reported significantly low errors, when compared to higher altitudes' measurements, that could be attributed to low GSD of 0.17 cm/pixel and high-resolution 3D reconstructed model. Practically, the overall range of errors at all flight altitudes falls within acceptable range that could provide sufficient information to conduct traffic accident investigations. However, the results suggest that lower altitudes with low GSD values would lead towards more accurate measurements of the accident scene.

**Table 5.** Accuracy indicators of scenario 1.

| Criteria | Flight Altitude | | | |
|---|---|---|---|---|
| | 17 m | 25 m | 30 m | 40 m |
| MSE | 0.95 | 3.01 | 9.17 | 13.68 |
| RMSE | 0.97 | 1.73 | 3.03 | 3.70 |
| CVRMSE | 1.72 | 3.06 | 5.30 | 6.61 |

### 4.2. Scenario 2: Variation of Frontal and Lateral Overlaps

In this section, we investigate the impact of varying percentage frontal and lateral overlaps on the accuracy of measurements. Four different combinations of overlap values were utilized in this investigation that were defined as:

- Case 1: Front overlap = 90% Side overlap 85%;
- Case 2: Front overlap = 85% Side overlap 80%;

- Case 3: Front overlap = 80% Side overlap 75%;
- Case 4: Front overlap = 75% Side overlap 65%.

The flight altitude was defined to be constant at 17 m during all the four cases. Experiments were run and data were gathered by sUAV and uploaded to Pix4D cloud platform. Four 3D models were constructed for each case and segments of interest were measured and presented in Table 6. The error percentages between the manually measured segment and the sUAV measured segment are illustrated in Figure 6.

**Table 6.** Manual measurements vs. sUAV measurements of the accident scene with varying percentages of frontal and lateral overlaps.

| Segment | Manual Measurements | Front 90% Side 90% | | Front 85% Side 80% | | Front 80% Side 75% | | Front 75% Side 65% | |
| --- | --- | --- | --- | --- | --- | --- | --- | --- | --- |
| | | sUAV (cm) | Error % | sUAV (cm) | Error % | sUAV (cm) | Error% | sUAV (cm) | Error% |
| C11 | 85.00 | 83.73 | 1.50 | 81.94 | 3.59 | 72.29 | 14.96 | 71.81 | 15.51 |
| C12 | 48.00 | 49.20 | 2.50 | 50.95 | 6.15 | 43.20 | 9.99 | 32.66 | 31.95 |
| C13 | 108.00 | 106.75 | 1.16 | 106.88 | 1.04 | 96.90 | 10.28 | 89.83 | 16.82 |
| C14 | 79.00 | 78.20 | 1.01 | 77.53 | 1.86 | 72.18 | 8.63 | 71.11 | 9.99 |
| C15 | 155.00 | 156.98 | 1.28 | 157.18 | 1.41 | 147.23 | 5.01 | 141.02 | 9.02 |
| C16 | 75.00 | 76.88 | 2.51 | 74.89 | 0.15 | 66.16 | 11.78 | 63.35 | 15.53 |
| C17 | 114.00 | 112.03 | 1.73 | 113.78 | 0.19 | 106.03 | 6.99 | 95.49 | 16.24 |
| C21 | 60.00 | 59.10 | 1.50 | 57.99 | 3.36 | 56.66 | 5.56 | 54.93 | 8.45 |
| C22 | 42.00 | 42.89 | 2.12 | 44.67 | 6.35 | 36.91 | 12.12 | 26.25 | 37.49 |
| C23 | 105.00 | 104.83 | 0.16 | 106.66 | 1.58 | 96.78 | 7.83 | 88.16 | 16.03 |
| C24 | 125.00 | 122.30 | 2.16 | 121.86 | 2.51 | 113.98 | 8.82 | 102.43 | 18.06 |
| C25 | 121.00 | 119.57 | 1.18 | 119.19 | 1.49 | 109.37 | 9.61 | 101.05 | 16.49 |
| C26 | 84.00 | 85.26 | 1.50 | 86.11 | 2.51 | 83.27 | 0.87 | 68.28 | 18.72 |
| C27 | 152.00 | 150.92 | 0.71 | 150.67 | 0.87 | 140.75 | 7.40 | 132.00 | 13.16 |
| A1 | 289.00 | 294.00 | 1.73 | 295.93 | 2.40 | 287.81 | 0.41 | 273.73 | 5.28 |
| A2 | 133.00 | 131.36 | 1.23 | 132.47 | 0.40 | 123.80 | 6.92 | 109.44 | 17.72 |
| A3 | 95.00 | 95.80 | 0.84 | 93.80 | 1.26 | 84.78 | 10.75 | 84.21 | 11.36 |

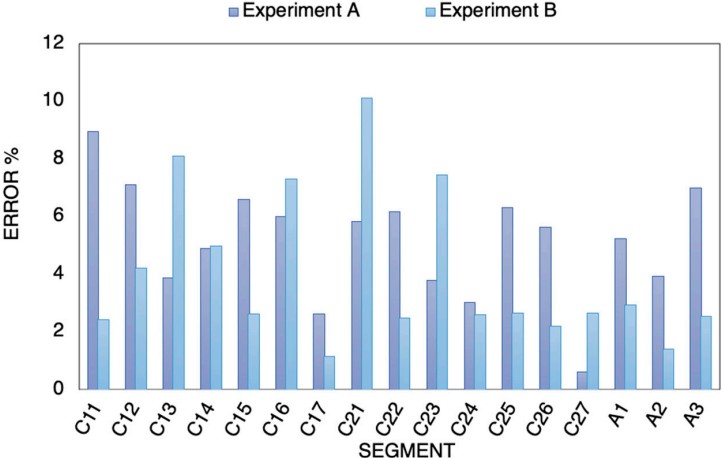

**Figure 6.** Error percentage between manual measurements and sUAV measured segments of Experiments A and B.

As illustrated in Figure 7, the accuracy of measurements deteriorates considerably when reducing the overlapping percentages. Statistical accuracy indicators for each case are presented in Table 7. It can be observed that the RMSE ranges between 1.86 and 16.26 with a significant increase that

negatively impacts the precision of measurements. Additionally, MAPE values range between 1.20% and 16.08%. Obviously, reducing the overlap percentages reduces the number of 3D point clouds and leads to a poorly reconstructed 3D model with missing information. With reference to the accuracy indicators of Table 5, cases 1 and 2 show acceptable range of errors for the application of road traffic reconstruction. The results therefore suggest that a minimum of 85% and 80% for frontal and lateral overlaps, respectively, is recommended to provide accurate 3D reconstructed traffic accident model.

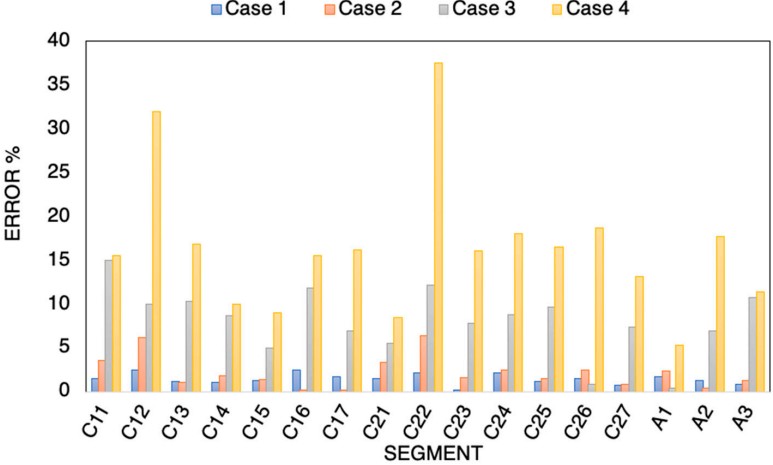

**Figure 7.** Error percent between manual measurements and sUAV measured segments with varying frontal and lateral overlaps.

**Table 7.** Accuracy indicators of scenario 2.

| Criteria | Frontal and Lateral Percent Overlap | | | |
|---|---|---|---|---|
| | **Front 90% Side 85%** | **Front 85% Side 80%** | **Front 80% Side 75%** | **Front 75% Side 65%** |
| MSE | 3.45 | 6.43 | 72.66 | 264.23 |
| RMSE | 1.86 | 2.54 | 8.52 | 16.26 |
| CVRMSE | 3.23 | 4.38 | 14.91 | 29.39 |
| MAPE | 1.20% | 6.10% | 8.22% | 16.08% |

*4.3. Scenario 3: Accuracy Assessment at Extreme Operating Conditions*

Experiments of Table 1 were conducted to assess the accuracy of the sUAV photogrammetry measurements at various operating conditions. To further examine the feasibility of adopting sUAV for photogrammetry of traffic accidents, it is crucial to test the system under extreme operating and environmental conditions of Kuwait. Therefore, experiment A conducted in July, which is often the hottest month of the year in Kuwait, with high temperature and dusty high winds. In addition, experiment B was conducted in November that corresponds to the other extreme of winter season with low temperature, low light, medium wind and scattered rain.

Experiments were run at a constant altitude of 17 m. Table 8 presents a comparison between the manual measurements and the acquired images by the reconstructed 3D models of each experiment. A comparison between the two experiments' error percentages is illustrated in Figure 7. At both experiments, error fluctuates between a minimum of 1.16% and a maximum of 10.11%. Table 9 provides the accuracy indicators of both experiments. It can be noticed that experiment A resulted in better measurements of the segments with a MAPE of 5.15% and RMSE of 4.66 that reflects the ability of sUAV to resist high winds, due to its design, and provide an acceptable error range within the extreme operating condition of the experiment. Moreover, the high temperature did not affect the process of the sUAV in acquiring the images of the scene. This is due to the short flight time during the low altitude flight mission and the dust that scatters the exposure of sunlight.

**Table 8.** Manual measurements vs. sUAV measurements of the accident scene in extreme operating conditions of Kuwait.

| Segment | Manual Measurements | Experiment A | | Experiment B | |
|---|---|---|---|---|---|
| | | sUAV (cm) | Error % | sUAV (cm) | Error % |
| C11 | 85.00 | 92.61 | 8.96 | 82.93 | 2.43 |
| C12 | 48.00 | 44.59 | 7.11 | 45.97 | 4.22 |
| C13 | 108.00 | 112.20 | 3.89 | 116.74 | 8.09 |
| C14 | 79.00 | 75.13 | 4.90 | 82.94 | 4.99 |
| C15 | 155.00 | 144.78 | 6.59 | 150.94 | 2.62 |
| C16 | 75.00 | 70.51 | 5.99 | 69.52 | 7.30 |
| C17 | 114.00 | 111.00 | 2.63 | 115.32 | 1.16 |
| C21 | 60.00 | 56.50 | 5.83 | 53.93 | 10.11 |
| C22 | 42.00 | 39.41 | 6.16 | 43.04 | 2.48 |
| C23 | 105.00 | 108.96 | 3.78 | 97.19 | 7.43 |
| C24 | 125.00 | 128.76 | 3.01 | 121.75 | 2.60 |
| C25 | 121.00 | 113.37 | 6.31 | 124.22 | 2.66 |
| C26 | 84.00 | 79.28 | 5.62 | 82.16 | 2.19 |
| C27 | 152.00 | 151.07 | 0.61 | 156.02 | 2.65 |
| A1 | 289.00 | 273.86 | 5.24 | 280.54 | 2.93 |
| A2 | 133.00 | 138.24 | 3.94 | 131.13 | 1.41 |
| A3 | 95.00 | 88.37 | 6.98 | 92.59 | 2.54 |

**Table 9.** Accuracy indicators of scenario 3.

| Criteria | Experiments | |
|---|---|---|
| | A | B |
| MSE | 21.70 | 39.22 |
| RMSE | 4.66 | 6.26 |
| CVRMSE | 8.35 | 11.50 |
| MAPE | 5.15% | 20.20% |

On the other hand, experiment B has a significantly high error percentage with a MAPE of 20.20% and RMSE of 6.26. The low light and scattered rain drops were the most challenging factors of experiment B that affected the quality of the reconstructed 3D model of the scene. This is due to the underexposed images with missing information that affects the build of 3D point clouds. One possible solution to this problem is to use portable lighting for the scene and incorporate the process of capturing high dynamic range (HDR) photos to reveal maximum information of each image. Another possible solution is to use a low-light lens to maximize the information at each captured image.

## 5. Discussion

The present study presented an accuracy assessment of a reconstructed traffic accident scene using sUAV operating at extreme operating conditions. Three experiments were conducted at different flight altitudes during various weather conditions including high temperature, low temperature, dusty wind gusts, rain and low light. Encouraging results, in terms of precision, were presented that range from highly accurate to fairly accurate measurements. In comparison with manual accident scene measurements, the utilization of sUAV and SfM in reconstructing the accident scene significantly reduce the time of investigation to reopen the roads and safer for police officers to gather all

the required information from the accident scene. In addition, the proposed process provides gathering detailed and informative evidences from the scene that can be documented and accessed in future for further analyses.

In this section, we validate the applicability of further analyses using complex real traffic accident scene with challenging non-elastic deformations, lighting and colors. Furthermore, we present various software tools and procedures that would enrich the expert's investigation process. In addition, limitations of the proposed framework are presented with suggestions for future research directions.

### 5.1. Applicability

Data collection procedure is the most imperative part of the traffic accident investigation. Detailed data are of the utmost importance in determining the faults, raising awareness in terms of possible road design and traffic routing flaws. Such information enriches the expert investigative procedure and quantifies the severity of the vehicle damages for insurance companies. Utilizing 3D modelling software packages such as Fusion360 (Autodesk Inc., San Rafael, CA, USA) or Blender (Blender, Amsterdam, The Netherlands) with the reconstructed 3D model of the accident scene could reveal important information about the incident.

A significant aspect of the inspection procedure is to simulate the accident. One possible procedure is to utilize physics engine-based simulation software such as Blender (Blender, The Netherlands) or Cinema4D (Maxon GmbH, Essen, Germany), to simulate the accident with y generating a 3D mesh object of the vehicles from the reconstructed 3D model of the vehicles. The experts can simulate the accident and determine the collision forces, collision angles, vehicle path and provide further complex simulations.

Furthermore, generating 3D mesh objects of the vehicle can be utilized to measure the volume of the deformed parts of the vehicle to provide a percentage of the damage for experts and insurance companies. The mesh surfaces can be generated by the photogrammetry software packages and then exported into 3D modelling software for further precise measurements.

Figure 8 presents a 3D model of a collided vehicle from a real two-cars traffic accident in Kuwait at Alrabiya area. The vehicle is a white four-wheel drive Nissan Patrol that was reconstructed using sUAV and the model was exported into Blender 3D modelling software for measurements. Figure 9 illustrate the generated 3D mesh objects from the reconstructed 3D model of the vehicle in Blender software. Using Blender inspection and measuring tools, volumes can be calculated and the deformed parts can be estimated, for example, as a percentage of the whole vehicle's volume. In Figure 9a, the volume of the whole vehicle was calculated as 19.8 m$^3$ while the deformation volume shown in Figure 9b was estimated as 1.52 m$^3$.

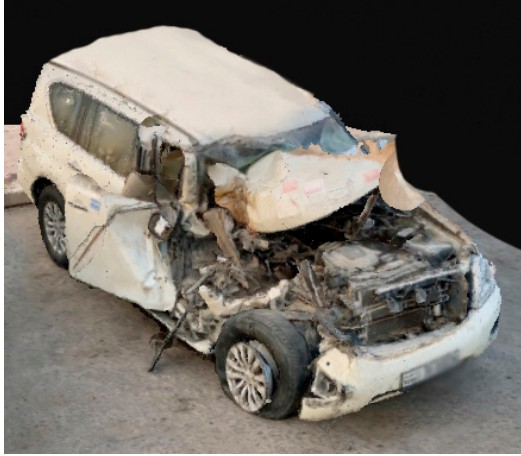

**Figure 8.** Photogrammetrically reconstructed 3D model of a real collided vehicle in Kuwait in the Alrabiya area.

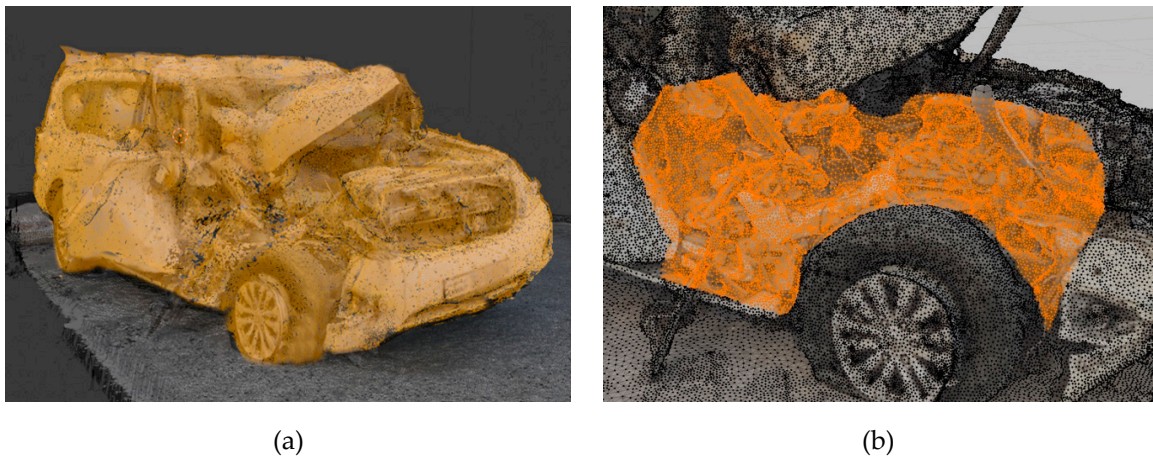

(a)                                                                              (b)

**Figure 9.** A generated 3D mesh objects of the vehicle 3D model in Blender (**a**) measuring the volume of the whole vehicle body (**b**) measuring the deformation volume of the collided frontal part of the vehicle.

Another significant aspect of utilizing 3D modelling software is that the damage profile of the vehicle can be defined by overlaying an undamaged 3D model of the vehicle to estimate the deformed parts volumes. Figure 10 illustrates a 3D reconstructed model of a white Ford Mustang car that collided with another vehicle in an accident at Alrabiya area of Kuwait as well as a photo of the real car from a handheld camera for a comparison. The deformations can be clearly noted at the front and side parts of the vehicle. The generated 3D mesh object can be then exported into Blender software and overlayed with an undamaged 3D model to define the areas of deformation and draw the damage profile as shown in Figure 11 in terms of 3D mesh vertices or edges.

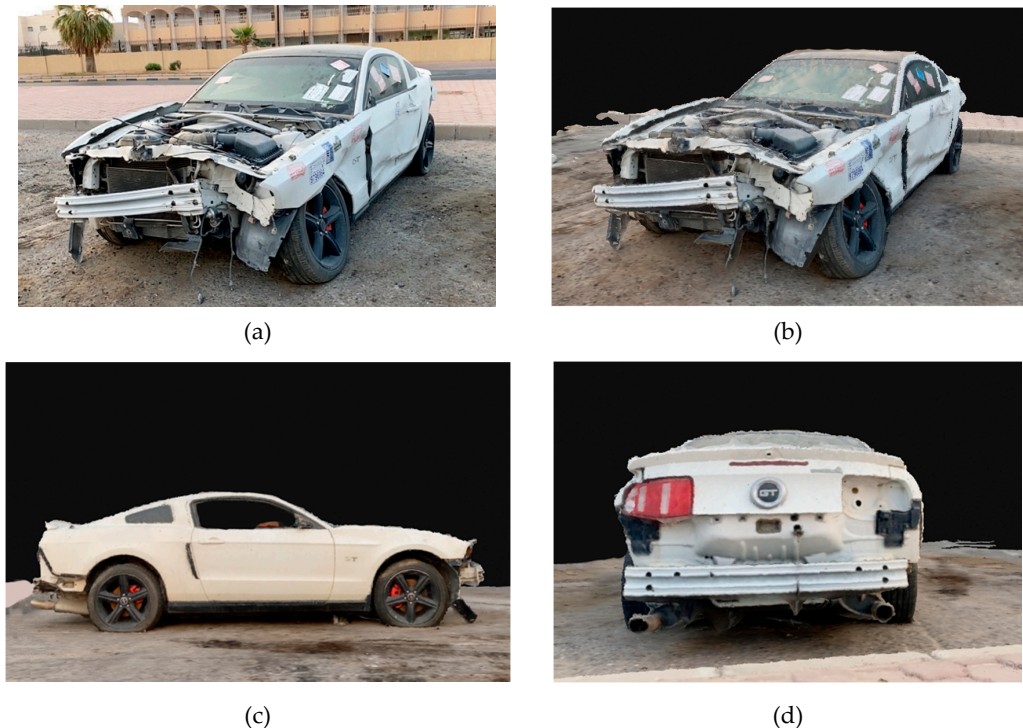

(a)                                                                              (b)

(c)                                                                              (d)

**Figure 10.** Photogrammetrically generated 3D model of a white Ford mustang vehicle from a two-car accident at Alrabiya area in Kuwait (**a**) photo from a handheld camera (**b**) 3D reconstructed model from sUAV from front left view (**c**) side view of the 3D model of the vehicle (**d**) back view of the 3D model of the vehicle.

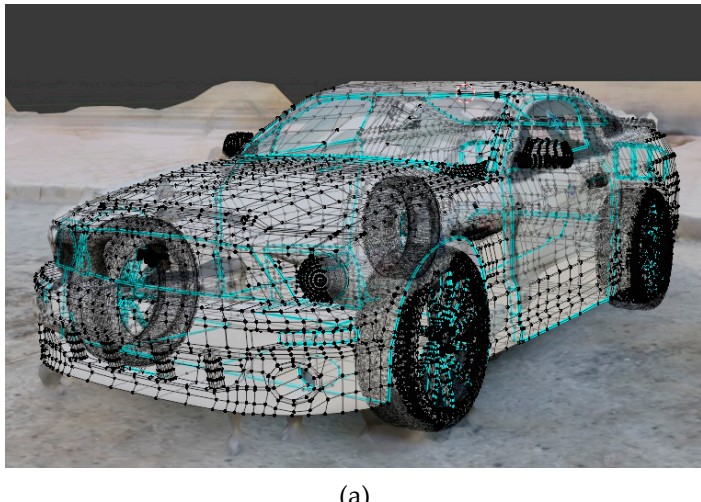 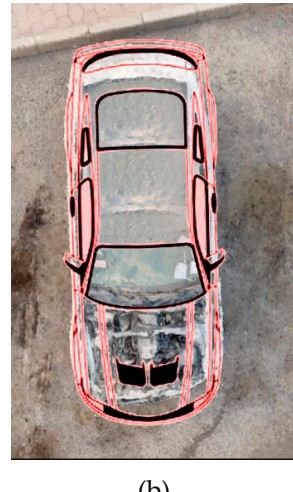

(a)                                                                                                    (b)

**Figure 11.** Damage profiles of the vehicle defined in terms of (**a**) 3D mesh vertices from front-left view (**b**) 3D mesh edges from a top view to assess the damage and deformation of the vehicle's body.

The demonstrated applicability of representing the damage profile as well as measuring the volume of deformed parts using the aforementioned procedures allow experts to gather information that was difficult to extract using the manual measurement techniques and significantly reduces the investigation procedure in an efficient workflow. In addition, the concluded information can be used generate reports for investigators, court cases and insurance companies with required accident measurements.

*5.2. Challenges, Limitations and Future Research Directions*

Despite the encouraging results and promising applicability that were demonstrated in previous sections, the generalizability of using sUAV photogrammetry framework is subject to some challenges and limitations. One major limitation arises when the accident occurs within a hard-to-fly area such as tunnels and areas with dense barriers such as forests. Similarly, there are restricted and critical-infrastructure areas where drones are prohibited such airports, petrochemical refineries, oil rigs and military bases. Handheld LiDAR may be used at these areas to capture the 3D point clouds of the scene and reconstruct the models. One major drawback of the LiDAR is that it captures high point cloud density that is more computationally expensive to reconstruct when compared with the proposed sUAV photogrammetry approach [25]. LiDAR also requires enhanced GPS and users need to be trained for photogrammetry techniques such as overlapping and angles to ensure capturing sufficient details of the scene to produce the 3D model. The LiDAR sensors can be integrated with sUAV but could increase the payload on sUAVs and decreases the flight time [25].

On the other hand, poor illuminations of the accident scene may lower the quality of the reconstructed 3D model [26]. In addition, reconstructing vehicles with reflective body paints, such as white and silver, are prone to unfavorable illuminations that degrades the quality of the reconstructed 3D model. A possible solution to overcome this challenge is by adjusting the camera settings to expose for the brightest areas of the scene and capture images using HDR mode to capture all the possible details of the scene [26]. This approach was carried out to build the vehicles shown in Figures 8 and 10. Another possible solution is to incorporate the use of artificial portable light sources at the accident scene to have a well illuminated scene for reconstruction however this approach would add more complexity and increase the investigation time.

Notwithstanding these limitations and challenges, the reported results attests the applicability and feasibility in adopting sUAV photogrammetry framework for accident scene measurements and documentation with a high extent of accuracy at extreme conditions. Further research to explore the integration of artificial intelligence algorithms to define optimal flight paths and the overlap

percentage would result in high quality reconstruction of the accident scene. Moreover, the integration of fully autonomous flight mission planning with all flight and image capture parameters could enhance the user experience and makes it easier for non-technical personnel to operate the system. Additionally, for an increased security and controlled processing of the data, edge computing platforms could be explored instead of cloud processing.

## 6. Conclusions

This study contributes to the existing knowledge of using sUAV in photogrammetry applications by exploring the measurement accuracy and robustness of the sUAV in extreme environments. The investigation of the performance of sUAV-based photogrammetry of traffic accidents at extreme operating environments has been presented. Simulated traffic accident scenes have been captured at different seasons of the year with the presence of various environmental factors such as low light, dust, high wind and high temperature. A low-cost sUAV was used to acquire the images from various flight altitudes and percentage overlaps between frontal and lateral consecutive images. Investigations on optimal operating parameters at normal and extreme operating conditions have been highlighted by the results of various experiments. Quantitative analyses were carried out to compare the actual measurements with the measurements from the reconstructed 3D model. The proposed method reported highly accurate to fairly accurate measurements of the scene at normal and extreme operating conditions, respectively. Results with RMSE range between 0.97 and 4.66 and a MAPE between 1.03% and 20.2%, that correspond to normal and extreme environments, respectively, have been successfully achieved and presented. Further experiments were demonstrated to show the feasibility of the approach to reconstruct accidents with complex deformations of the vehicles. The applicability of the approach was reviewed in terms of how experts can benefit from the 3D reconstructed outputs to provide an in-depth analysis and complex measurements. The reported results provide a significant insight on the robustness of sUAV in reconstructing traffic accidents 3D models at extreme operating conditions. However, this approach is not applicable at fly-restricted zones and hard to access areas such as tunnels and areas with severe obstructions. The findings encourage the adoption of low-cost sUAVs in reconstructing the accident scene as a permanent feature due to the reported feasibility at normal and extreme environments; therefore, motivating the authorities to adopt the presented framework to eventually replace the current classical accident measurement approach and enhance the accident investigation procedures.

**Author Contributions:** Conceptualization, A.M.A. and M.R.A.; methodology, A.M.A. and M.R.A.; software, A.M.A.; validation, A.M.A., M.R.A., and A.K.A.; formal analysis, A.M.A. and A.K.A. investigation, A.M.A. and M.R.A.; resources, A.M.A., A.K.A., data curation, A.M.A. and M.R.A.; writing—original draft preparation, A.M.A., M.R.A., M.R.A., and A.K.A.; writing—review and editing, A.M.A., M.R.A. and A.K.A.; visualization, A.M.A., M.R.A. and A.K.A.; All authors have read and agreed to the published version of the manuscript.

**Funding:** This research received no external funding.

**Acknowledgments:** This research was conducted at the robotics research lab of the electronics engineering department, college of technological studies, PAAET, Kuwait.

**Conflicts of Interest:** The authors declare no conflict of interest.

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
