# Peer review of "Accuracy Assessment of Small Unmanned Aerial Vehicle for Traffic Accident Photogrammetry in the Extreme Operating Conditions of Kuwait"

_information, doi:10.3390/info11090442_

Round 1

Reviewer 1 Report

The paper provides an interesting concept of using 3D reconstruction to analyze traffic accidents. Although it sounds interesting, the authors must ensure the efficiency of their approach in real-like scenarios.

  • In case of a real collision, the cars would suffer a nonelastic deformation. How good is their work for that? 
  • Some colors are difficult to construct, such as silver and white cars.
  • More in-depth detail about precision and applicability is necessary. For instance, how can an expert use this reconstruction to conclude better than just pictures?
  • Photometry is also highly sensitive to illumination changes, how to avoid that problem?

In my opinion, the paper's goal is excellent and important, i.e., to have 3D models for remote inspection. However, much of the paper was dedicated to well-known results, and the main idea was not well explored. For instance, it is possible that a US$ 1.000,00 ground lidar would provide better results in terms of details, quality, security, applicability, and several other advantages. I highly recommend testing this different approach.

In resume, the idea and problem are very important, but the results must explore more complex scenarios, the paper should focus more on the final analysis and relevance of the approach than the trivial methodology of how those point clouds were obtained. 

Author Response

Dear respected reviewer, 

Please find attached the response to your kind comments. 

Many thanks, 

Regards, 

Authors

Reviewer 2 Report

MS: Accuracy Assessment of Small Unmanned Aerial Vehicle for Traffic Accident Photogrammetry in the Extreme Operating Conditions of Kuwait

The study examines the accuracy of traffic accident using sUAV photogrammetry. The accuracy assessment of accident scene was compared using manual and sUAV measurements under three different scenarios.

Positive – The study is novel and used three scenarios to assess accuracy

Negative – Some results in the Tables are replicated in Figures.

No discussion and Conclusion missing the main points

Generally, sUAV coupling with Structure-from-Motion but no any information provided

Title: Not sure that the geographic region is required for the title. If not change ‘of Kuwait’ to ‘in Kuwait’

L 31-32: Of 4.27 million populations, approximately 185,000 road ….

Authors use UAVs and sUAV interchangeably. Select one term and used throughout.

L 151: space ( Bentley

L 152: space ( DroneDeploy

Figure 4. Orthomosaic and DSM images of the scene.

Table 4: Remove bold C11

Figure 6 represent the error values in Table 4, which replicate same results. Instead of replicating Fig. 6 the authors can bold some figures in table 4 to highlight key findings.

Figure 7: Use distinctive color ramp for different patterns in gray scale

Discussion- No discussion given

Conclusion: Take-home message missing in the conclusions. What conditions are the best for assessing traffic accident?

Author Response

(The authors gave the same response as above.)

Round 2

Reviewer 1 Report

congratulations.

Author Response

Dear respected reviewer,

Thank you for your efforts in reviewing our manuscript and the constructive comments that have substantially improved the article.

Kind regards,

Authors

Reviewer 2 Report

The authors made good inputs for the manuscript to improve its contents.

However, I have a few comments for further improvement of manuscript.

L 122: Specify the software Agisoft Metashape, Pix4Dmapper, ArcGISPro etc.

L 31-32: I mean restructure the sentence (i.e. Of 4.27 million populations, approximately 185,000 road traffic accidents were recorded in Kuwait between 2016-2019 as per the Central Statistical Bureau of Kuwait (CSB) [3]) not addition the populations after figure.

Good to see the discussion section in the manuscript. However, only one reference [17] used for the discussion, which typically a gap of the discussion. Literature citation for the manuscript is lean

Author Response

Dear respected reviewer,

Thank you for the time you dedicated in reviewing our manuscript and we greatly appreciate your insightful comments that helped us in improving the manuscript. Herewith attached, we provide a point-by-point response to your kind comments.

Regards, 

Authors
